# The Impacts of Antivirals on the Coronavirus Genome Structure and Subsequent Pathogenicity, Virus Fitness and Antiviral Design

**DOI:** 10.3390/biomedicines8100376

**Published:** 2020-09-24

**Authors:** Ching-Hung Lin, Cheng-Yao Yang, Shan-Chia Ou, Meilin Wang, Chen-Yu Lo, Tsung-Lin Tsai, Hung-Yi Wu

**Affiliations:** 1Graduate Institute of Veterinary Pathobiology, College of Veterinary Medicine, National Chung Hsing University, Taichung 40227, Taiwan; tw23whale@hotmail.com (C.-H.L.); yangchengyao@nchu.edu.tw (C.-Y.Y.); axfbji7917@gmail.com (C.-Y.L.); windtaker10@msn.com (T.-L.T.); 2Graduate Institute of Microbiology and Public Health, College of Veterinary Medicine, National Chung Hsing University, Taichung 40227, Taiwan; scou@dragon.nchu.edu.tw; 3Department of Microbiology and Immunology, Chung Shan Medical University, Taichung 40201, Taiwan; wml@csmu.edu.tw

**Keywords:** coronavirus, genome structure, pathogenicity, remdesivir, innate immunity, antiviral drug, spike protein

## Abstract

With the global threat of SARS-CoV-2, much effort has been focused on treatment and disease control. However, how coronaviruses react to the treatments and whether the surviving viruses have altered their characteristics are also unanswered questions with medical importance. To this end, bovine coronavirus (BCoV), which is in the same genus as SARS-CoV-2, was used as a test model and the findings were as follows. With the treatment of antiviral remdesivir, the selected BCoV variant with an altered genome structure developed resistance, but its pathogenicity was not increased in comparison to that of wild type (wt) BCoV. Under the selection pressure of innate immunity, the genome structure was also altered; however, neither resistance developed nor pathogenicity increased for the selected BCoV variant. Furthermore, both selected BCoV variants showed a better efficiency in adapting to alternative host cells than wt BCoV. In addition, the previously unidentified feature that the spike protein was a common target for mutations under different antiviral treatments might pose a problem for vaccine development because spike protein is a common target for antibody and vaccine designs. The findings derived from this fundamental research may contribute to the disease control and treatments against coronaviruses, including SARS-CoV-2.

## 1. Introduction

Coronaviruses (CoVs) are single-stranded, positive-sense RNA viruses with a genome size of 26–32 kilobases (kb) and belong to the subfamily *Coronavirinae*, family *Coronaviridae*, order *Nidovirales* [1,2,3,4]. Based on the serological evidence and genome structure, the subfamily *Coronavirinae* is divided into four genera: *Alphacoronavirus*, *Betacoronavirus*, *Gammacoronavirus*, and *Deltacoronavirus*. Among the seven known human coronaviruses (HCoVs), HCoV-OC43, SARS-CoV-1, MERS-CoV, and SARS-CoV-2 belong to *Betacoronavirus*, whereas HCoV-229E and HCoV-NL63 belong to *Alphacoronavirus* [3]. The genome structure of the coronavirus consists of a 5′-untranslated region (UTR), open reading frames (ORFs), and a 3′-UTR including a poly(A) tail. The nonstructural proteins (nsps) are encoded from the 5′ two-thirds of the genome; the other one-third of the genome contains mostly structural protein gene [2,5]. In addition to the genome, a nested set of subgenomic mRNAs (sgmRNAs) are also produced from which the structural proteins are synthesized during the coronavirus infection [6].

Since the outbreak of severe acute respiratory syndrome (SARS) [7], Middle East respiratory syndrome (MERS) [8], and novel coronavirus pneumonia (officially called COVID-19) [9], several approved drugs and investigational agents have been tested for the treatment of these diseases [10]. Of these antivirals, a nucleotide analogue prodrug remdesivir (GS-5734) has inhibitory effects on coronaviruses in cell cultures and in animal models, including feline infectious peritonitis virus (FIPV), mouse hepatitis virus A59 (MHV-A59), SARS-CoV-1, MERS-CoV, and SARS-CoV-2 [11,12,13,14]. The clinical trials in humans suggest that remdesivir is able to shorten the time to recovery in adults with COVID-19 and lower respiratory tract infection in comparison with placebo [15]. It has been demonstrated that in HIV-1, hepatitis C virus, and influenza A virus (IAV), under antiviral selection, viruses evolve to develop drug resistance by alterations of genome structure such as mutations in replicase genes, allowing viral populations that are not completely eliminated to repopulate within a host and thus posing a public health issue [11,16,17,18,19,20]. In HIV-1, hepatitis B virus, and hepatitis C virus, resistance mutations are usually selectively advantageous under the selection environment of antivirals; however, in the absence of antivirals, such resistance mutations may lose their advantage, which is usually attributed to mutation-associated fitness costs [21,22,23]. Thus, mutations with greater resistance in the aforementioned viruses may display higher fitness costs in the absence of selection pressure [24]. In IAV H1N1, oseltamivir-resistant H1N1 with mutation H275Y is generally emerged in immunocompromised patients [25,26,27]; however, the resistance mutation caused by antiviral oseltamivir in influenza A virus H1N1 has been found to have little or no fitness costs in the absence of oseltamivir possibly due to compensatory mutations [28,29]. The mechanism by which coronavirus resistance develops has been less extensively explored in coronaviruses, and the effects followed by the development of resistance on the subsequent treatment and disease control remain to be elucidated especially at the current stage of COVID-19 pandemic.

Viruses are constantly under the pressure of immune surveillance. Of the antiviral immune responses, innate immunity, which involves nonspecific immune defense mechanisms, is the very first line of cell defense against virus infection [30]. To establish an efficient infection under the constant pressure of innate immunity, viruses may evolve alterations in their genome structure that may affect the subsequent pathogenesis and virulence. In adaptive immunity, neutralization antibodies are mainly directed against the viral exposed surface proteins. It has been found that the hemagglutinin genome structure of IAV has been altered in order to escape adaptive immunity [31,32,33]. Such strategy is also employed by hepatitis C virus for efficient infection in which the glycoprotein E2 gene is also altered under the selective pressure of neutralizing antibodies [34,35]. In poliovirus, optimization of the genetic composition has been shown to be a strategy to overcome the challenges posed by innate immunity [36]. Whether coronavirus employs a similar strategy to withstand the constant selection pressure of innate immunity remains unknown.

Because of the lack of proofreading capability, the RNA-dependent RNA polymerase (RdRp) encoded by RNA virus has high mutation rates during RNA synthesis, leading to a diverse population of viruses or quasispecies [37,38]. Accordingly, the genetic structure of RNA virus populations is often depicted as a network of variants organized in sequence space around a single master sequence [39,40]. When faced with environmental challenges such as persistence, antivirals, immune responses, and new hosts, if only couple of mutations are required for the virus populations to rapidly reach an adapted genotype, the virus possesses genetic robustness by definition [41]. In contrast, if many mutations from the populations are required to arrive at an optimal genotype, the virus is genetically less robust. In this case, mutations in the gene structure of virus populations may act as stepping stones toward the synthesis of a new genotype to adapt to a new environment [42]. It has been suggested that RNA viruses with genetic robustness allow the viral population to explore an extensive region of sequence space, leading to numerous individuals that can rapidly withstand the environmental changes [43]. In an animal experiment of poliovirus, the wild-type (wt) virus population, which can adapt to environmental change with less time and shows more virulent than its variant containing synonymous mutations, is thus genetically robust [44]. In coronavirus, whether the wt coronavirus is more genetically robust than its variants remains to be determined and may have important implications in disease development and control.

With the global threat of SARS-CoV-2, much effort has been focused on treatments and prevention for the disease; however, how coronaviruses react to the treatments and whether the surviving viruses have altered their characteristics such as pathogenicity after weathering different pressures are also important questions that remain unanswered. Because bovine coronavirus (BCoV) (i) has been intensively used in replication study of coronavirus [45,46,47], (ii) shows higher sequence identity with HCoVs HCoV-OC43, HCoV-HKU1, SARS-CoV-1, MERS-CoV, and SARS-CoV-2 (*Betacoronavirus*) than HCoVs HCoV-229E and HCoV-NL63 (*Alphacoronavirus*) based on the phylogenetic tree analysis [3] and (iii) also belongs to genus *Betacoronavirus* [3], it was therefore selected as a test model to explore the impacts of antivirals including remdesivir and innate immunity on the genome structure and subsequent pathogenicity and virus fitness. The results derived from the study may contribute to medically important antiviral designs and clinical treatments against coronavirus infection.

## 2. Experimental Section

### 2.1. Viruses and Cells

Human rectum tumor (HRT)-18 cells [48] and Mouse L (ML) cells were obtained from David A. Brian (University of Tennessee, Knoxville, TN, USA) and maintained in Dulbecco’s modified Eagle’s medium (DMEM) supplemented with 10% fetal bovine serum (Hyclone, Logan, UT, USA) at 37 °C with 5% CO_2_. Bovine coronavirus (BCoV) strain of Mebus (GenBank accession no. U00735), which was also obtained from David A. Brian, was plaque-purified [49,50] and grown in HRT-18 cells.

### 2.2. Treatments of Cells with Antivirals

For selection pressure with antiviral remdesivir (GS-5734), HRT-18 cells were infected with wild type (wt) BCoV at a multiplicity of infection (MOI) of 1 and after 1 h of infection, HRT-18 cells were treated with GS-5734 with final concentration of 10 μM based on the study showing that the antiviral activity of GS-5734 at this concentration is able to inhibit replication of SARS-CoV-1 and MERS-CoV [11]. This stage of infection was defined as virus passage 0 (VP0). The virus at VP0 was then collected at 48 h postinfection (hpi) and then passaged to fresh HRT-18 cells (defined as VP1) in the presence of GS-5734 with final concentration of 10 μM [11]. To mimic the study in which 10-day dosing durations of the GS-5734 can shorten the time to recovery in adults infected with SARS-CoV-2 [15], viruses were passaged ten times (VP10) in HRT-18 cells in the presence of GS-5734. The virus collected from HRT-18 cells in the presence of GS-5734 at VP10 was designated WtGS10. The wt BCoV collected from HRT-18 cells in the absence of GS-5734 at VP10 was designated Wt10. To determine the effects of constant pressure of innate immunity on the genome structure and subsequent viral characteristics, HRT-18 cells were respectively transfected or mock-transfected with polyinosinic:polycytidylic acid (poly IC), a stimulant of innate immunity with a double-stranded RNA structure, with final concentration of 1 μg/mL [51]. After 4 h of treatment with poly IC, HRT-18 cells were infected with wt BCoV. The virus at VP0 was then collected at 48 hpi and then passaged to fresh HRT-18 cells (defined as VP1) in the presence of poly IC with final concentration of 1 μg/mL. The virus collected from HRT-18 cells in the presence of poly IC at VP10 was designated WtIC10. At each passage of virus, cellular RNA and lysates were collected for the subsequent assays. In addition, for the adaptation experiments of Wt10, WtGS10, and WtIC10, ML cells were respectively infected with the aforementioned virus and at VP10, cellular RNA and lysates were collected for subsequent assays.

### 2.3. Determination of Virus Titer

Plaque assay for determining the titer of BCoV was described previously [52]. In brief, BCoV with serial dilution was added into HRT-18 cells and at 1 hpi, HRT-18 cells were washed with DMEM followed by an agarose overlay containing DMEM, 0.6% agarose and 2% FBS. HRT-18 cells were then incubated at 37 °C with 5% CO_2_ for 72 h. Viral plaques were visualized by haemadsorption with mouse red blood cells. Blood was collected in Alsiever solution and centrifuged at low speed. The supernatant was discarded followed by washing with phosphate-buffered saline. The virus titer was determined by the number of haemadsorption foci.

### 2.4. Determination of Genome Structure

The terminal sequence of BCoV genomic RNA was identified using a head-to-tail ligation method as described previously [53]. The head-to-tail ligated RNA was used for reverse transcription (RT) with SuperScript III reverse transcriptase (Invitrogen, Carlsbad, CA, USA). PCR was performed with AccuPrime Taq DNA polymerase (Invitrogen, Carlsbad, CA, USA) and oligonucleotides binding to 5′ and 3′ UTR of the genome followed by sequencing. To determine the sequences of the genome, total cellular RNA was extracted with TRIzol (Invitrogen, Carlsbad, CA, USA) and random hexamer oligonucleotides were used for RT with SuperScript III reverse transcriptase (Invitrogen, Carlsbad, CA, USA), and the resulting cDNA was used for PCR with PfuUltra II high-fidelity DNA polymerase (Agilent, Santa Clara, CA, USA). The resultant PCR products were then subject to sequencing analysis.

### 2.5. Northern Blot Assay

Ten microgram of TRIzol-extracted total cellular RNA collected from the aforementioned experiments was electrophoresed through a formaldehyde-agarose gel for 3 h. By vacuum blotting, RNA was transferred from the gel to a Nytran membrane followed by UV cross-linking. BCoV RNA was detected with the oligonucleotide, which was 5′-end labeled with ^32^P and bound to BCoV 3′ UTR. The probed blot was exposed to Kodak XAR-5 film (Kodak, Rochester, NY, USA).

### 2.6. RT-qPCR

To determine the synthesis efficiency of BCoV RNA, 2 μg of TRIzol-extracted total cellular RNA was used for the RT reaction. To quantitate the synthesis of BCoV genome, oligonucleotides binding to the leader sequence and sequence in the nsp1 gene were used. For measurement of subgenome N synthesis, oligonucleotides binding to the leader sequence and sequence downstream of the start codon for N gene were used. For qPCR, SYBR^®^ green amplification mix (Roche Applied Science, Mannheim, Germany) and oligonucleotides were used according to the manufacturer’s protocol. In these experiments, dilutions of plasmids containing the same gene as the detected genome and subgenome were always run in parallel with the quantitated cDNA for use in standard curves (dilutions ranged from 10^8^ to 10 copies of each plasmid). The amount of synthesized RNA was normalized to the levels of internal control 18S rRNA.

### 2.7. Western Blot Assay

The harvested cell lysates were separated using 12% sodium dodecyl sulphate-polyacrylamide gel electrophoresis (SDS-PAGE) gels. After electrophoresis, samples were electrotransferred onto nitrocellulose membranes (GE Healthcare, Chicago, IL, USA). An antibody against BCoV nsp1, nucleocapsid (N) protein, or β-actin was used as the primary antibody followed by goat anti-mouse IgG conjugated to horseradish peroxidase (HRPO) as the secondary antibody (Jackson Laboratory, Bar Harbor, ME, USA). Detected protein(s) was visualized using Western Lightning™ Chemiluminescence Reagent (Perkin Elmer, Waltham, MA, USA) and Kodak XAR-5 film (Kodak, Rochester, NY, USA).

### 2.8. Statistical Analysis

Student’s *t*-test was used for the analysis of the data using Prism 6.0 software (GraphPad Software Inc., San Diego, CA, USA). The values in the study are presented as the means ± standard deviations (SD) (*n* = 3); * *p* < 0.05, ** *p* < 0.01 and *** *p* < 0.001.

## 3. Results

### 3.1. The Genome Structure of BCoV Is Altered under the Treatment of Remdesivir GS-5734 in HRT-18 Cells

Antiviral therapy with drug such as nucleos(t)ide analogues targeting the viral polymerase can inhibit viral replication and subsequent disease progression. Conversely, viruses may also undergo mutations for survival under the treatment of antivirals, leading to the emergence of drug resistance and treatment failure. To test whether the genome structure was altered under antiviral selection pressure, a series of wt BCoV passages in the presence of antiviral remdesivir GS-5734 were performed. The virus collected at virus passage (VP) 10 in the absence of GS-5734 was designated Wt10 while the virus collected in the presence of GS-5734 was designated WtGS10. In comparison with the genome sequence of wt BCoV (Wt), few nt mutations were identified but no nonsynonymous mutations were found in Wt10. However, as illustrated in Figure 1A, in comparison with the genome sequence of Wt10, multiple nt mutations in WtGS10 were identified and mostly occurred in genes of replication-associated protein and S protein. In addition, dispersed aa mutations were identified in the replication-associated proteins nsps 1, 3, 5, 6, 8, 9, 12, and 14 (Figure 1B) and multiple aa mutations (42 aa) occurred in S protein, with the majority (37 out of 42 aa) in the S1 subunit (Figure 1C). Thus, under the repeated treatment of antiviral GS-5734, the genome structure was altered and the altered sequences mostly occurred in the genes of replication-associated proteins and S protein.

### 3.2. After the Repeated Treatments of GS-5734, BCoV Develops Resistance but Its Pathogenicity Is Not Increased

To test whether GS-5734 had an inhibitory effect on the replication efficiency of wt BCoV (Wt) in HRT-18 cells, a plaque assay and Northern blot analysis were employed. The titer (10^7.54^ pfu/mL) of wt BCoV in the presence of GS-5734 (Wt + GS, Figure 2A) decreased by three-fold in comparison to that of wt BCoV (10^8.09^ pfu/mL) in the absence of GS-5734 (Wt, Figure 2A) in HRT-18 cells. The amounts of viral RNA (represented by subgenomes) in the presence of GS-5734 (Wt + GS) also decreased by ~30% in comparison to those in the absence of GS-5734 (Wt), as determined by Northern blotting (Figure 2B). Together, the results suggested that GS-5734 could inhibit viral synthesis. To test whether the altered genome structures (Figure 1) in wt BCoV after 10 passages in the presence of GS-5734 (i.e., WtGS10) maintained its replication efficiency in HRT-18 cells in the absence of GS-5734, a plaque assay was performed. As shown in Figure 2C, the virus titer of WtGS10 (10^6.89^ pfu/mL) was lower by ~13-fold in comparison to that of Wt10 (i.e., wt BCoV after 10 passages in the absence of GS-5734) (10^8.02^ pfu/mL). The amounts of viral RNA synthesis collected from WtGS10-infected HRT-18 cells were also decreased by ~30% in comparison to those from Wt10-infected HRT-18 cells in the absence of GS-5734 (Figure 2D). These results suggested that the replication efficiency of WtGS10 decreased in comparison to that of Wt10 in HRT-18 cells in the absence of GS-5734. That is, the pathogenicity of antiviral remdesivir-selected virus was not increased in comparison with that of wt BCoV in HRT-18 cells. To further test whether the resultant WtGS10 had developed resistance, HRT-18 cells were infected with WtGS10 in the absence or presence of GS-5734. After 24 h of infection, the virus from HRT-18 cells in the presence of GS-5734 was collected (designed WtGS10 + GS) and subjected to plaque assay. As shown in Figure 2C, the virus titer (10^7.44^ pfu/mL) of WtGS10 + GS (in the presence of GS-5734) was ~3-fold higher than that of WtGS10 (10^6.89^ pfu/mL, in the absence of GS-5734), but still ~four-fold lower than that of Wt10. The amounts of viral RNA synthesis of WtGS10 + GS were also increased by ~20% when compared with those of WtGS10 (Figure 2D). The results suggested that BCoV had developed resistance after the treatment of GS-5734 at VP10.

### 3.3. The Antiviral GS-5734-Selected WtGS10 Adapts to ML Cells with Better Efficiency than Wt BCoV (Wt10)

Since multiple aa substitutions occurred in the S protein of WtGS10 (Figure 1), it was also of importance to test whether such mutations have altered its characteristics such as adaptation capability to alternative host cells. The results may have important implications in interspecies transmission and disease control. To address this, ML cells, a mouse cell line established from subcutaneous connective tissue, were respectively infected with Wt10 and WtGS10. After 24 h of infection, the virus (VP0) was collected to infect fresh ML cells (VP1). The virus passage step was repeated until VP10, and Northern blot analysis was performed to detect the synthesis of viral RNA. As shown in Figure 3A, lanes 3 and 4, both viral RNAs were detected, suggesting WtGS10 could adapt to ML cells. However, the amounts of viral RNAs (as determined by Northern blot and RT-qPCR analyses, Figure 3A,B, respectively) and proteins (as determined by Western blotting, Figure 3C) in ML cells were overall higher for WtGS10 than Wt10, suggesting that the antiviral GS-5734-selected WtGS10 adapted to ML cells with better efficiency than wt BCoV (Wt10). Sequence analysis suggested that the majority of mutations occurred in the S1 subunit of S protein for both Wt10 and WtGS10 in ML cells at VP10 (Wt10-ML and WtGS10-ML, respectively, in Figure 3D). We also found that although both Wt10 and WtGS10 were able to adapt to ML cells, their S protein sequence showed lower identity (27-aa difference). Interestingly, the resulting S protein sequence of WtGS10 after adaptation to ML cells became very similar to that of Wt10 in HRT-18 cells (only 7-aa difference between WtGS10-ML and Wt10-HRT, but 26-aa difference between Wt10-ML and Wt10-HRT, Figure 3D).

Together, the results suggested that the repeated treatment of antiviral GS-5734 led to the synthesis of WtGS10 with drug resistance, possibly by alterations of the genome structure. Such alterations may contribute to virus fitness in the antiviral (GS-5734) environment, but may not be advantageous for increasing the pathogenicity of the virus in the absence of GS-5734 in HRT-18 cells. In addition, the WtGS10 selected by antiviral GS-5734 adapted to ML cells with better efficiency than wt BCoV (Wt10) and thus may have implication in interspecies transmission and disease control.

### 3.4. The Genome Structure of BCoV Is Altered in HRT-18 Cells Treated with Poly IC-Induced Innate Immunity

In addition to antiviral drug, whether the constant pressure of innate immunity could lead to the alterations of the genome structure and subsequent viral characteristics remained to be determined in coronaviruses. Thus, HRT-18 cells were respectively transfected or mock-transfected with poly IC, a stimulant of innate immunity with a double-stranded RNA structure, followed by wt BCoV infection. At VP10, the virus (designated WtIC10) and RNA were collected for subsequent analyses. After sequencing analysis, in comparison with the genome sequence of Wt10, multiple nt mutations in WtIC10 were identified and mostly occurred in genes of replication-associated protein and S protein (Figure 4A). In addition, dispersed aa substitutions were identified in replication-associated proteins, including nsps 1, 3, 5, 6, 8, 9, 12, and 14, and N protein (Figure 4B). Of the 26 aa mutations in S protein, 22 aa occurred in the S1 subunit (Figure 4C). Therefore, under selection pressure of poly IC-stimulated innate immunity, the genome structure of BCoV was also altered, and the altered aa sequences mostly occurred in replication-associated proteins and S protein.

### 3.5. Neither Resistance Develops nor Pathogenicity Increases in BCoV under the Treatment of Poly IC-Induced Innate Immunity

To test whether poly IC had an inhibitory effect on the replication of wt BCoV in HRT-18 cells, a plaque assay and Northern blot analysis were employed. The titer of wt BCoV (Wt) in the presence of poly IC (10^7.58^ pfu/mL, Wt + IC, Figure 5A) decreased by three-fold in comparison to that of wt BCoV in the absence of poly IC (10^8.09^ pfu/mL, Wt, Figure 5A). The amounts of RNA collected in the presence of poly IC (Wt + IC, Figure 5B) also decreased by ~38% when compared with those in the absence of poly IC (Wt, Figure 5B). These results suggested that innate immunity induced by poly IC had an inhibitory effect on the production of wt BCoV in HRT-18 cells. To test whether the alterations of genome structures (Figure 4) in WtIC10 (i.e., wt BCoV after 10 passages in the presence of poly IC) affected the replication efficiency in the absence of poly IC in HRT-18 cells, a plaque assay was performed. As shown in Figure 5C, the virus titer of WtIC10 (10^7.84^ pfu/mL) was lower by ~two-fold in comparison to that of Wt10 (i.e., wt BCoV after 10 passages in the absence of poly IC) (10^8.05^ pfu/mL). The amounts of viral RNA synthesis collected from WtIC10-infected HRT-18 cells were also decreased by ~30% in comparison to those collected from Wt10-infected HRT-18 cells in the absence of poly IC (Figure 5D). These results suggested that, in the absence of poly IC, the replication efficiency of WtIC10 was slightly decreased in comparison to that of Wt10 in HRT-18 cells. That is, the pathogenicity of the virus selected by poly IC-induced innate immunity is not increased in comparison with that of wt BCoV in HRT-18 cells. To test further whether the resultant WtIC10 has developed resistance, HRT-18 cells were treated with or without poly IC followed by infection with WtIC10. After 24 h of infection, the virus from HRT-18 cells treated with poly IC (designated WtIC10 + IC) or without poly IC (WtIC10) was collected and subjected to a plaque assay. As shown in Figure 5C, the virus titer was ~two-fold lower for WtIC10 + IC (10^7.48^ pfu/mL, in the presence of poly IC) than WtIC10 (10^7.84^ pfu/mL, in the absence of poly IC). The amounts of viral RNA statistically showed no significant difference between WtIC10 and WtIC10 + IC (Figure 5D). The results suggest that BCoV did not develop resistance against poly IC-induced innate immunity.

### 3.6. The WtIC10 Selected by Poly IC-Induced Innate Immunity Adapts to ML Cells with Better Efficiency than Wt BCoV (Wt10)

To address whether the multiple aa substitutions occurred in S protein of WtIC10 affected its adaptation capability to ML cells, ML cells were respectively infected with Wt10 and WtIC10. After 24 h of infection, the virus (VP0) was collected to infect fresh ML cells (VP1). At VP10, both viral RNAs collected from Wt10- and WtIC10-infected ML cells were detected (Figure 6A, lanes 3 and 4, respectively). In addition, the amounts of viral RNAs (as determined by Northern blotting and RT-qPCR analyses, Figure 6A,B, respectively) and proteins (as determined by Western blotting, Figure 6C) in ML cells were overall higher for WtIC10 than Wt10. These results suggest that the selected WtIC10 by poly IC-induced innate immunity could adapt to ML cells and its adaptation efficiency was superior to that of Wt10. Although both Wt10 and WtIC10 could adapt to ML cells at VP10, their S protein sequence between the two viruses showed low identity (26-aa difference) (Wt10-ML and WtIC10-ML, respectively, in Figure 6D). In addition, as with WtGS10, the resulting S protein sequence of WtIC10 after adaptation to ML cells became very similar to that of Wt10 in HRT-18 cells (only 4-aa difference between WtIC10-ML and Wt10-HRT but 24-aa difference between Wt10-ML and Wt10-HRT, Figure 6D).

These results together suggest that BCoV does not develop resistance and increased pathogenicity after the repeated treatment of poly IC-induced innate immunity in HRT-18 cells. However, as demonstrated with WtGS10 in Figure 3, the selected WtIC10 adapted to ML cells with better efficiency than wt BCoV.

## 4. Discussion

In this study, we employed BCoV, which is in the same genus (*Betacoronavirus*) as SARS-CoV-1, SARS-CoV-2, and MERS-CoV, as a test model to explore how coronavirus reacts to the treatments of antiviral remdesivir GS-5734 and poly IC-induced innate immunity in light of alterations of the genome structure and subsequent virus fitness. We anticipate that the results may have medical importance on pathogenicity, treatments, transmission, and antiviral design for coronavirus, and are discussed below.

After a series of passages of mouse hepatitis virus A59 (MHV-A59) in the presence of GS-5734 in DBT cells, 4 aa mutations in the replication-associated proteins and 2 aa mutations in S protein are found [11]. However, in the current study examining BCoV in HRT-18 cells, surprisingly, 31 aa mutations in replication-associated proteins, including nsps 1, 3, 5, 6, 8, 9, 12, and 14 (Figure 1B) and 42 aa mutations in S protein were identified (Figure 1C). It is rational to suggest that the mutations occurred in replication-associated proteins because they are potential targets for antiviral GS-5734, although the number of mutations in HRT-18 cells is higher (31 aa, Figure 1B) than that in a previous report using DBT cells with MHV-A59 (4 aa) [11]. However, the occurrence of multiple mutations in S protein (42 aa) is unusual. Based on the available data [11] and current study (Figure 1C), we reason that different coronaviruses and host cells may lead to different outcomes under the pressure of antiviral GS-5734 regarding the mutations in S protein. Consequently, the mutations in S protein selected under antiviral GS-5734 in the current and previous study [11] may provide clues to gain insights into the antiviral mechanism of GS-5734.

In terms of function-orientated selection, the selected mutations for WtIC10 were expected to be associated with resistance to poly IC treatment. However, the mutations in WtIC10 showed almost no effect on replication efficiency (WtIC10 vs WtIC10 + IC in Figure 6C,D) in the presence of poly IC, suggesting that poly IC-derived innate immunity may not be a strong selection pressure for BCoV in HRT-18 cells. Because innate immunity is the first line of cellular defense against virus infection and viruses also have to constantly withstand the challenge for survival, it is argued that evolutionarily wt BCoV genome may have evolved strategies to counteract the innate immune response for successful infection. Thus, the altered genome structures in this study may not aim to withstand the pressure of poly IC-induced innate immunity and may have other unidentified functions.

In the experiment of adaptation capability in ML cells, it is surprising that, after adapting to ML cells, the resulting S protein aa sequences between WtGS10 and WtIC10 (WtGS10-ML and WtIC10-ML, respectively in Appendix AA) showed high identity (only 4-aa difference) although their S protein aa sequences showed low identity (24-aa difference) before adapting to ML cells (i.e., WtGS10 and WtIC10 in HRT-18 cells; WtGS10-HRT and WtIC10-HRT in Appendix AB). We reason that, because of constant mutation and selection, each of the selected BCoV variants under the antiviral pressures (in HRT-18 cells) contains a different network of neutral mutations that can act as stepping-stones toward the synthesis of a similar genotype to adapt to the same host cells (ML cells) or same receptor, resulting in the similar S protein aa sequences. In line with this proposition, the network of variants in the populations of WtGS10 and WtIC10 may have more neutral mutations in adapting to ML cells (or possibly the receptor) than those of Wt10 to facilitate the synthesis of a new genotype for adaptation, leading to a better capability to adapt to new host cells than wt BCoV (Figure 3 and Figure 6) and thus genetically more robust. In addition, the adaptation results also raise another question of why, after adapting to ML cells, the aa sequence identity of S protein between Wt10-ML and WtGS10-ML (26-aa difference, Appendix AA) and between Wt10-ML and WtIC10-ML (22-aa difference, Appendix AA) is low, but they all can adapt to ML cells efficiently (although the adaptation efficiency is better for WtGS10 and WtIC10 than Wt10, Figure 3 and Figure 6)? We speculate that the species in the network of mutations for Wt10 are different from those for WtGS10 and WtIC10 in HRT-18 cells (Appendix AB) and consequently, Wt10 in ML cells may evolve in a direction different from that of WtGS10 and WtIC10 in order to rapidly reach a new genotype for binding to an alternative receptor different from that of WtGS10 and WtIC10, leading to a different S protein aa sequence. Interestingly, we also found that the resulting S protein aa sequences of the variants WtGS10 and WtIC10 at VP10 (in ML cells) showed high identity with those of Wt10 (in HRT-18 cells) (7-aa difference between WtGS10-ML and Wt10-HRT; 4-aa difference between WtIC10-ML and Wt10-HRT, Appendix AA). Although further study is required to elucidate the mechanism leading to these findings, the features may pose an issue regarding the transmission of coronaviruses between different host species. Together, this study suggests that the genome structure in the network of variants in the coronavirus populations has a profound influence on the adaptation capability, and thus, the viral fitness and pathogenicity. In this sense, genetic robustness of coronavirus can facilitate its adaptation and thus evolution [42,44].

It has been suggested that adaptive immunity can lead to mutations in viral surface proteins [31,32,33,34,35]. In infectious bronchitis virus (IBV), an avian coronavirus, it has been demonstrated that the adaptive immunity caused by vaccination can lead to mutations of coronavirus S protein, especially the S1 subunit, the main domain responsible for binding to the cellular receptor and antibody [55]. It has also been shown that S protein is associated with protective immunity [56] and that mutations in S protein can lead to the failure of protective immunity [57,58]. Thus, mutations in S protein are always the main concern during vaccine development and disease control. In this study, we did not expect BCoV S protein to represent the same preferential target for selected mutations of antivirals including remdesivir GS-5734 and poly IC-induced innate immunity tested in this study in light of function-orientated selection. It remains unclear why different pressures led to genome alterations with a higher mutation intensity at the same target (S protein). One explanation may be that, evolutionarily, in order to infect different host cells for survival, the S protein gene in wt coronavirus has evolved to contain hotspot mutation sequences in the virus population during infection and led to sequence diversity, thus rapidly adapting to environmental challenges. Consequently, although the resultants WtGS10 and WtIC10 which were selected under different selection pressures did not develop increased pathogenicity, the high proportion of mutations in the S protein in these selected mutants may have implications in vaccine development and subsequent disease control.

Taken together, we in this study employed BCoV as a test model to explore how coronavirus reacted to the treatments of antiviral remdesivir GS-5734 and poly IC-induced innate immunity in terms of the genome structure, pathogenicity, and adaptation capability. The obtained results from this study may have implications in treatments and antiviral design for coronavirus. However, further in vitro and in vivo studies using SARS-CoV-2 are required to validate the current results and thus can contribute to the treatments of coronavirus infection, including SARS-CoV-2.

## 5. Conclusions

In conclusion, with the treatment of antiviral remdesivir, the selected BCoV variant with an altered genome structure develops resistance, but its pathogenicity is not increased in comparison to that of wild type (wt) BCoV. In contrast, under the selection by poly IC-induced innate immunity, the genome structure is also altered; however, neither resistance develops nor pathogenicity increases for the selected BCoV variant. Furthermore, both selected BCoV variants show a better efficiency in adapting to alternative host cells than wt BCoV, highlighting the diverse features of coronavirus genome. Thus, the results may provide information contributing to disease control and treatments against coronavirus infection.

## Figures and Tables

**Figure 1 biomedicines-08-00376-f001:**
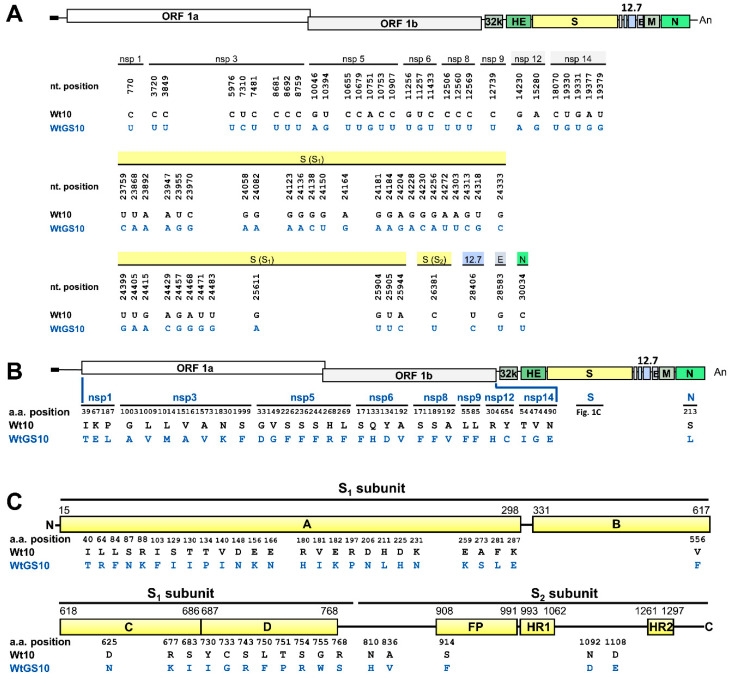
Linear schematic of BCoV genome showing the location of mutated nt and aa under the treatment of GS-5734 in HRT-18 cells. (**A**) Linear schematic of BCoV genome showing the comparison of the nt sequences identified from HRT-18 cells infected with wt BCoV (Wt) in the presence (WtGS10) or absence (Wt10) of GS-5734 at VP10. (**B**) Linear schematic of BCoV genome showing the comparison of the aa sequences identified from HRT-18 cells infected with Wt in the presence (WtGS10) or absence (Wt10) of GS-5734 at VP10. (**C**) Linear schematic of BCoV S protein with subunits S1 (domains A–D) and S2 (FP, HR1, and HR2) showing the positions of the aa alterations identified from HRT-18 cells infected with Wt in the presence (WtGS10) or absence (Wt10) of GS-5734 at VP10. The residue numbering is based on the BCoV Mebus strain spike protein (GenBank: U00735) with domain boundaries based on the HCoV-OC43 S structure, Adapted from Hulswit, R.J.G.; Lang, Y.F.; Bakkers, M.J.G.; Li, W.T.; Li, Z.S.; Schouten, A.; Ophorst, B.; van Kuppeveld, F.J.M.; Boons, G.J.; Bosch, B.J.; et al. Human coronaviruses OC43 and HKU1 bind to 9-O-acetylated sialic acids via a conserved receptor-binding site in spike protein domain A. *Proc. Natl. Acad. Sci. USA*
**2019**, *116*, 2681–2690 [54]. ORF: open reading frame, 32K: 32 kDa protein, HE: hemagglutinin/esterase, S: spike protein, 12.7: 12.7 kDa protein, E: envelope protein, M: membrane protein, N: nucleocapsid protein, FP: fusion peptide, HR1: heptad repeat 1, HR2: heptad repeat 2.

**Figure 2 biomedicines-08-00376-f002:**
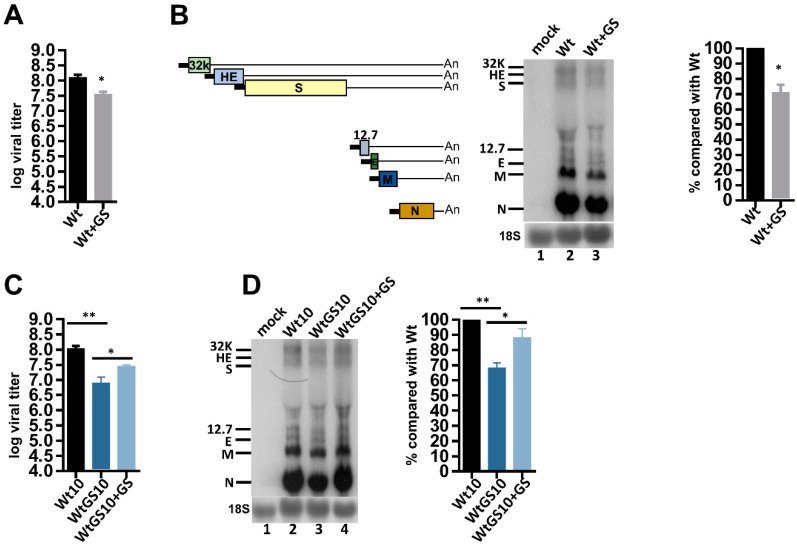
Effects of antiviral remdesivir (GS-5734) on the virus fitness in HRT-18 cells. (**A**) The virus titer of Wt and Wt + GS at VP0 as determined by the plaque assay. Wt indicates the virus collected from wt BCoV-infected HRT-18 cells in the absence of GS-5734. Wt + GS indicates the virus collected from wt BCoV-infected HRT-18 cells in the presence of GS-5734. (**B**) Left panel: Schematic representing the BCoV sgmRNAs. Middle panel: determination of BCoV RNA synthesis from HRT-18 cells infected with wt BCoV in the absence (Wt) or presence (Wt + GS) of GS-5734 at VP0 by Northern blotting. BCoV subgenomes are used to represent the RNA synthesis because the signal of genome is generally weak or not detectable using Northern blotting. 32K: 32 kDa protein, HE: hemagglutinin/esterase, S: spike protein, 12.7: 12.7 kDa protein, E: envelope protein, M: membrane protein, N: nucleocapsid protein. Right panel: Relative amounts of BCoV RNA between Wt and Wt + GS based on the results shown in the middle panel. (**C**) Comparison of the viral titer between viruses collected from Wt10- and WtGS10-infected HRT-18 cells in the absence of GS-5734, and between viruses collected from WtGS10-infected HRT-18 cells in the absence of GS-5734 (i.e., WtGS10) and from WtGS10-infected HRT-18 cells in the presence of GS-5734 (i.e., WtGS10 + GS), as determined by the plaque assay. Wt10 is the virus collected from wt BCoV-infected HRT-18 cells at VP10 in the absence of GS-5734. WtGS10 is the virus collected from wt BCoV-infected HRT-18 cells at VP10 in the presence of GS-5734. (**D**) Left panel: determination of BCoV RNA synthesis from Wt10- and WtGS10-infected HRT-18 cells in the absence of GS-5734 (lanes 2 and 3, respectively), and from WtGS10-infected HRT-18 cells in the presence of GS-5734 (WtGS10 + GS, lane 4) by Northern blotting. Right panel: relative amounts of BCoV RNA based on the results shown in the left panel. 32K: 32 kDa protein, HE: hemagglutinin/esterase, S: spike protein, 12.7: 12.7 kDa protein, E: envelope protein, M: membrane protein, N: nucleocapsid protein, 18S: 18S rRNA, FP: fusion peptide, HR1: heptad repeat 1, HR2: heptad repeat 2. The values in (**A**–**D**) represent the mean ± standard deviation (SD) of three individual experiments. The statistical significance was evaluated using a paired *t*-test: * *p* < 0.05, ** *p* < 0.01.

**Figure 3 biomedicines-08-00376-f003:**
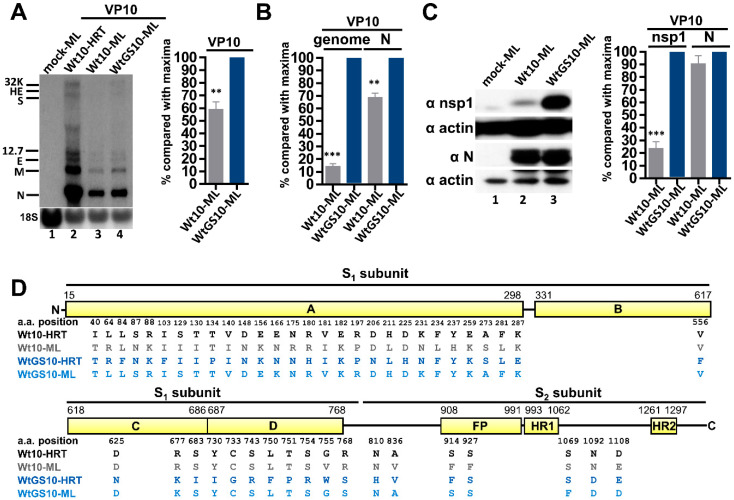
Adaptation capability of the remdesivir-selected BCoV variant WtGS10 in ML cells. (**A**) Left panel: determination of BCoV RNA synthesis from HRT-18 infected with Wt (Wt10-HRT, lane 2), or from ML cells infected with Wt10 (Wt10-ML, lane 3) or WtGS10 (WtGS10-ML, lane 4) at VP10 by Northern blotting. Right panel: Relative amounts of BCoV RNA from ML cells infected with Wt10 or WtGS10 at VP10 based on the results shown in the left panel. (**B**) Relative amounts of genome and subgenome (represented by subgenome N) from ML cells infected with Wt10 or WtGS10 at VP10 as measured by RT-qPCR. (**C**) Left panel: Coronavirus protein synthesis from the genome (represented by nsp1) and subgenome (represented by N protein) from ML cells infected with Wt10 or WtGS10 at VP10 by Western blotting. Right panel: Relative amounts of nsp1 and N protein between Wt10 and WtGS10 based on the results shown in the left panel. (**D**) Linear schematic of BCoV S protein showing the comparison of the aa sequences identified from HRT-18 cells infected with Wt at VP10 (Wt10-HRT, i.e., Wt10 in Figure 1), ML cells infected with Wt at VP10 (Wt10-ML), HRT-18 cells infected with Wt in the presence of GS-5734 at VP10 (WtGS10-HRT, i.e., WtGS10 in Figure 1) and ML cells infected with WtGS10 at VP10 (WtGS10-ML). 32K: 32 kDa protein, HE: hemagglutinin/esterase, S: spike protein, 12.7: 12.7 kDa protein, E: envelope protein, M: membrane protein, N: nucleocapsid protein, 18S: 18S rRNA, FP: fusion peptide, HR1: heptad repeat 1, HR2: heptad repeat 2. The values in (**A**–**C**) represent the mean ± standard deviation (SD) of three individual experiments. The statistical significance was evaluated using an unpaired *t*-test: ** *p* < 0.01, *** *p* < 0.001.

**Figure 4 biomedicines-08-00376-f004:**
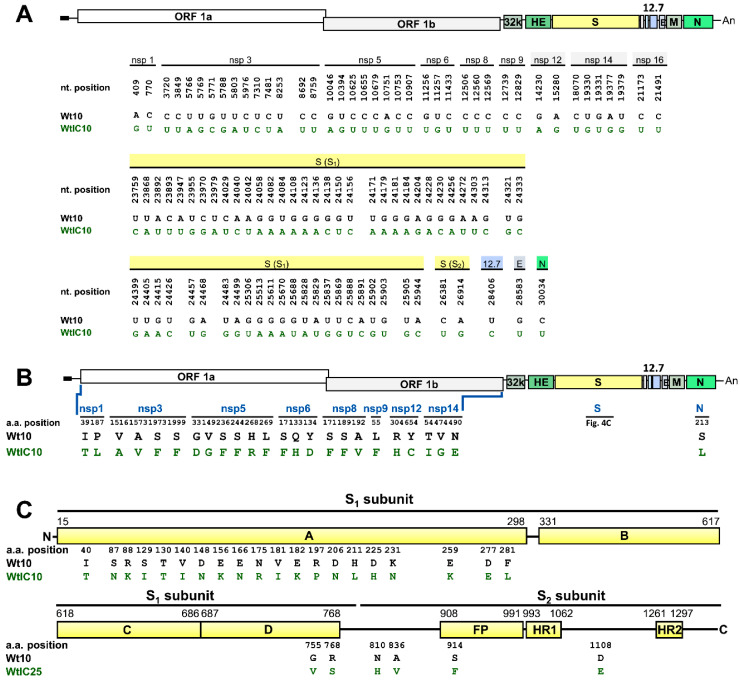
Linear schematic of BCoV genome showing the location of mutated nt and aa caused by poly IC-induced innate immunity in HRT-18 cells. (**A**) Linear schematic of BCoV genome showing the comparison of the nt sequences identified from HRT-18 cells infected with wt BCoV (Wt) in the presence (WtIC10) or absence (Wt10) of poly IC at VP10. (**B**) Linear schematic of BCoV genome showing the comparison of the aa sequences identified from HRT-18 cells infected with Wt in the presence (WtIC10) or absence (Wt10) of poly IC at VP10. (**C**) Linear schematic of BCoV S protein with subunits S1 (domains A–D) and S2 (FP, HR1, and HR2) showing the positions of the aa alterations identified from HRT-18 cells infected with Wt in the presence (WtIC10) or absence (Wt10) of poly IC at VP10. ORF: open reading frame, 32K: 32 kDa protein, HE: hemagglutinin/esterase, S: spike protein, 12.7: 12.7 kDa protein, E: envelope protein, M: membrane protein, N: nucleocapsid protein, FP: fusion peptide, HR1: heptad repeat 1, HR2: heptad repeat 2.

**Figure 5 biomedicines-08-00376-f005:**
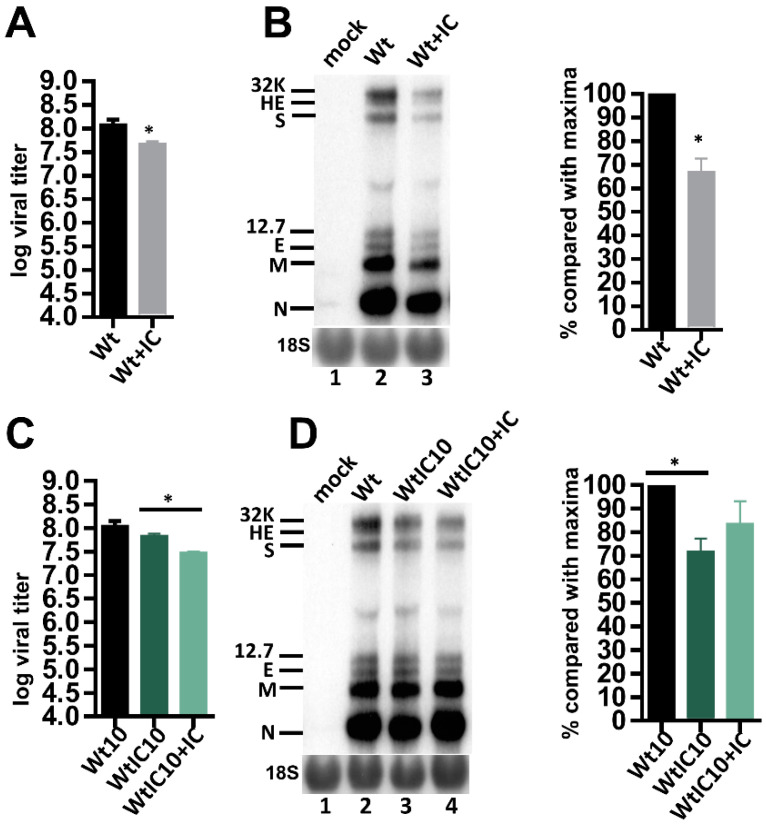
Effects of poly IC-induced innate immunity on the virus fitness in HRT-18 cells. (**A**) The viral titer of Wt and Wt + IC at VP0 as determined by the plaque assay. Wt indicates the virus collected from wt BCoV-infected HRT-18 cells in the absence of poly IC at VP0. Wt + IC indicates the virus collected from wt BCoV-infected HRT-18 cells in the presence of poly IC at VP0. (**B**) Left panel: determination of BCoV RNA synthesis from HRT-18 cells infected with wt BCoV in the absence (Wt) or presence (Wt + IC) of poly IC at VP0 by Northern blotting. Right panel: relative amounts of BCoV RNA between Wt and Wt + IC based on the results shown in the left panel. (**C**) Comparison of the viral titer between viruses collected from Wt10- and WtIC10-infected HRT-18 cells in the absence of poly IC, and between viruses collected from WtIC10-infected HRT-18 cells in the absence of poly IC (i.e., WtIC10) and from WtIC10-infected HRT-18 cells in the presence of poly IC (i.e., WtIC10 + IC), as determined by the plaque assay. Wt10 is the virus collected from wt BCoV-infected HRT-18 cells at VP10 in the absence of poly IC. WtIC10 is the virus collected from wt BCoV-infected HRT-18 cells at VP10 in the presence of poly IC. (**D**) Left panel: determination of BCoV RNA synthesis from Wt10- and WtIC10-infected HRT-18 cells in the absence of poly IC (lanes 2 and 3, respectively) and from WtIC10-infected HRT-18 cells in the presence of poly IC (WtIC10 + IC, lane 4) by Northern blotting. Right panel: relative amounts of BCoV RNA based on the results shown in the left panel. 32K: 32 kDa protein, HE: hemagglutinin/esterase, S: spike protein, 12.7: 12.7 kDa protein, E: envelope protein, M: membrane protein, N: nucleocapsid protein, 18S: 18S rRNA. The values in (**A**–**D**) represent the mean ± standard deviation (SD) of three individual experiments. The statistical significance was evaluated using a paired *t*-test: * *p* < 0.05.

**Figure 6 biomedicines-08-00376-f006:**
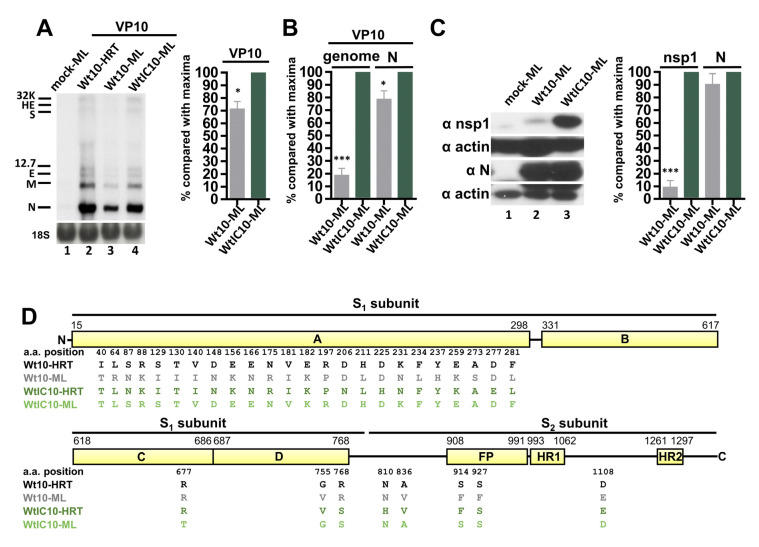
Adaptation capability of the selected BCoV variant WtIC10 in ML cells. (**A**) Left panel: determination of BCoV RNA synthesis from HRT-18 infected with Wt (Wt10-HRT, lane 2) or from ML cells infected with Wt10 (Wt10-ML, lane 3) or WtIC10 (WtIC10-ML, lane 4) at VP10 by Northern blotting. Right panel: relative amounts of BCoV RNA from ML cells infected with Wt10 or WtIC10 at VP10 based on the results shown in the left panel. (**B**) Relative amounts of genome and subgenome (represented by sgmRNA N) from fresh ML cells infected with Wt10 or WtIC10 at VP10 as measured by RT-qPCR. (**C**) Left panel: coronavirus protein synthesis from the genome (represented by nsp1) and subgenome (represented by N protein) from ML cells infected with Wt10 or WtIC10 at VP10 by Western blotting. Right panel: relative amounts of nsp1 and N protein between Wt10 and WtIC10 based on the results shown in the left panel. (**D**) Linear schematic of BCoV S protein showing the comparison of the aa sequences identified from HRT-18 cells infected with Wt at VP10 (Wt10-HRT, i.e., Wt10 in Figure 4), ML cells infected with Wt at VP10 (Wt10-ML), HRT-18 cells infected with Wt in the presence of poly IC at VP10 (WtIC10-HRT, i.e., WtIC10 in Figure 4) and ML cells infected with WtIC10 at VP10 (WtIC10-ML). 32K: 32 kDa protein, HE: hemagglutinin/esterase, S: spike protein, 12.7: 12.7 kDa protein, E: envelope protein, M: membrane protein, N: nucleocapsid protein, 18S: 18S rRNA, FP: fusion peptide, HR1: heptad repeat 1, HR2: heptad repeat 2. The values in (**A**–**C**) represent the mean ± standard deviation (SD) of three individual experiments. Statistical significance was evaluated using an unpaired *t*-test: * *p* < 0.05, *** *p* < 0.001.

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
