# Peer review of "The Impacts of Antivirals on the Coronavirus Genome Structure and Subsequent Pathogenicity, Virus Fitness and Antiviral Design"

_biomedicines, 2020, doi:10.3390/biomedicines8100376_

Round 1

Reviewer 1 Report

The manuscript ‘The impacts of antivirals on the coronavirus genome structure and subsequent pathogenicity, virus fitness and antiviral design’ by Ching-Hung Lin et al. has been reviewed.

The paper describes the efforts made by authors to study the development of BCoV resistant variants after the treatment with the antiviral drug remdesivir and poly IC-induced innate immunity.

The manuscript is of interest, it is well written and the literature is adequate, however since BCoV was used in the study as a model of SARS-CoV-2 infection the discussion should be rewritten to make more clear to the readers that additional studies are necessary to demonstrate that the obtained results are useful for SARS-COV-2 therapy.

The manuscript presents several imprecision and typos, that should be revised before publication, few examples are reported below:

-Page 3 line 10 “48 hour”, page 4 line 2 “the the”, page 5 line 14, page 6 line 4 (mantained and plague assay), page 7 line 15 and 16,  page 8 line 20, page 10 line 1, page 11 line 21, page 13 line 7.

-The literature should be carefully revised and the style conformed.

In my opinion, the reported work should be accepted in biomedicines after minor revision noted.

Author Response

Dear Reviewer:

We thank you for the detailed reviews and the valuable comments for improvement of our manuscript. We have responded to the comments with details, in particular the spelling errors and Discussion section presented in the previous manuscript, as follows.

Reviewer #1

  1. The manuscript ‘The impacts of antivirals on the coronavirus genome structure and subsequent pathogenicity, virus fitness and antiviral design’ by Ching-Hung Lin et al. has been reviewed.
  2. The paper describes the efforts made by authors to study the development of BCoV resistant variants after the treatment with the antiviral drug remdesivir and poly IC-induced innate immunity.
  3. The manuscript is of interest, it is well written and the literature is adequate, however since BCoV was used in the study as a model of SARS-CoV-2 infection the discussion should be rewritten to make more clear to the readers that additional studies are necessary to demonstrate that the obtained results are useful for SARS-COV-2 therapy.

Authors’ response:

We thank the reviewer for the valuable suggestion to make the manuscript more clear. We have rewritten the Discussion section by emphasizing that further studies are required to validate  the current results and thus can contribute to the SARS-CoV-2 therapy. This has been addressed on page 14, lines 24-29.

  1. The manuscript presents several imprecision and typos, that should be revised before publication, few examples are reported below:

Page 3 line 10 “48 hour”, page 4 line 2 “the the”, , page 5 line 14, page 6 line 4 (mantained and plague assay), , page 7 line 15 and 16,  page 8 line 20, page 10 line 1, page 11 line 21, page 13 line 7.

Authors’ response:

In addition to the correction of the imprecision and typos below, we also have corrected other errors in the revised manuscript.

Page 3 line 10 “48 hour”: We have corrected this (48 hours) on page 3, line 28.

page 4 line 2 “the the”: We have deleted additional “the” on page 4, line 23.

page 5 line 14: We have corrected this (develops) on page 6, line 11.

page 6 line 4 (mantained and plague assay): We have corrected these (maintained and plaque assay) on page 6, line 21.

page 7 line 15 and 16: We have corrected these (characteristics and alternative) on page 7, lines 25-26.

page 8 line 20: We have corrected this (characteristics) on page 9, lines 9.

page 10 line 1: We have corrected this (plaque) on page 10, lines 18.

page 11 line 21: We have corrected this (develop) on page 12, lines 6.

page 13 line 7: We have corrected this (network) on page 13, lines 36.

  1. The literature should be carefully revised and the style conformed.

Authors’ response:

We thank the reviewer for point out the inconsistent style of literature. We have revised the literature and conformed the style. We also have deleted the inconsistent style “(6,7)” on page 2, line 2 in the original manuscript. 

In my opinion, the reported work should be accepted in biomedicines after minor revision noted.

Sincerely,

Hung-Yi Wu, D.V.M., M.S., Ph.D.

Professor of Graduate Institute of Veterinary Pathobiology

College of Veterinary Medicine

National Chung Hsing University

Tel:886-4-22840369

Fax:886-4-22862073

Email:[email protected]

Reviewer 2 Report

Overall:

The Reviewer would like to thank the authors for the opportunity to review this interesting and timely manuscript. This manuscript represents a good effort to approach a topic of merit in the COVID-19 arena. Below are the reviewer’s comment which could hopefully help the authors in improving their manuscript. The overall feeling of the Reviewer is that although the paper is well-designed and elegantly written, its in vitro experiments are somehow preliminary (besides, these concepts are known from other viruses and antiviral drugs) and would benefit from further experimentation that is already available within the COVID-19 literature. Doing so would, in turn assist the authors in making their claims in the Discussion and Conclusions sections as more substantive. Of course, the final option is left to the Authors and the final discretion to the Editors of the manuscript and the Journal.

Comments:

  1. Please, provide your ORCID comments.
  2. 2, line 3-4: Please, provide data from clinical trials (or even better meta-analyses published in leading medical journals) on this drug.
  3. 2, line 11: Please, comment how often is this the case in immunocompromised patients, e.g., see relevant discussion in Influenza by using appropriate citations such as Eur J Clin Microbiol Infect Dis. 2017 Feb;36(2):361-371. doi: 10.1007/s10096-016-2809-
  4. Pg 2., line 19: The reader may wonder to know what happens in adaptive immunity, as well.
  5. 2, line 44: Please, elaborate on information about the homology of bovine coronavirus with human coronaviruses.
  6. 3, line 10: Please, explain how this final concentration was selected.
  7. 3, line 13: Please, explain how studying innate immunity is possible within in vitro settings? Please, also, elaborate on what is meant by “treatment of innate immunity”? So, after having read the Results sections as well, the reviewer would like to request to please explain early on that this can be performed using, for example, stimulants of innate immunity with a double-stranded RNA structure.
  8. 3, line 22: Since the comparison is made between “before” and “after” treatment, why was an unpaired t-test, and not paired t-test applied?
  9. 4, line 32: The phrase “[…] performed based on the clinical trial results” appears unsupported from the Methods section. Please, provide explanations.
  10. 12, line 32 (Optional): Would it be possible to perform these experiments at the authors’ side?
  11. 12, line 21-41 (Optional): This text is rather speculative. Although elegantly written, the reviewer recommend that the authors either “tone it down” or create a figure and state these elements as “hypothesis”, or even better perform some (at least preliminary yet publication-standards in vitro experiments) to justify their conclusions.
  12. 12, line 21-41: Please, conduct at least some molecular dynamics experiments to justify your claims, following similar examples in the field: https://pubmed.ncbi.nlm.nih.gov/32637944/
  13. 13-14, “Conclusions” section: Please, “tone down” these statements.
  14. Pg 13, line 30: Please, comment on whether there are experiments to assess your findings in connection to the adaptive immunity.

Minor comments:

  1. 3, line 8: Please, provide explanation of the acronym MOI.
  2. 4, line 22: Please, provide the name of the city for the company that produces the software.
  3. 7, line 16: Please, correct typo to “alternative host cells”.
  4. 11, line 21: Please, correct typo to “develope”.
  5. 13, line 28: Please, insert commas as follows: “capability, and, thus, the viral fitness …”
  6. 13, line 38: Please, correct typo to “evolutionarily”.
  7. General comment: Please, do not separate the Figures from their legends in separate lines, as the reader’s attention is heavily dispersed.

Author Response

Dear Reviewer:

We thank you for the detailed reviews and the valuable comments for improvement of our manuscript. We have responded to the comments with details, in particular the Discussion and Conclusions sections presented in the previous manuscript, as follows.

Reviewer #2

Overall

The Reviewer would like to thank the authors for the opportunity to review this interesting and timely manuscript. This manuscript represents a good effort to approach a topic of merit in the COVID-19 arena. Below are the reviewer’s comment which could hopefully help the authors in improving their manuscript. The overall feeling of the Reviewer is that although the paper is well-designed and elegantly written, its in vitro experiments are somehow preliminary (besides, these concepts are known from other viruses and antiviral drugs) and would benefit from further experimentation that is already available within the COVID-19 literature. Doing so would, in turn assist the authors in making their claims in the Discussion and Conclusions sections as more substantive. Of course, the final option is left to the Authors and the final discretion to the Editors of the manuscript and the Journal. 

Comments:

1. Please, provide your ORCID comments

Authors’ response:

ORCID iD is as follows: https://orcid.org/0000-0002-1260-6259

(please let me know if I need to provide further information, thanks)

2. Page 2, line 3-4: Please, provide data from clinical trials (or even better meta-analyses published in leading medical journals) on this drug.

Authors’ response:

The current data from clinical trials on this drug suggest that GS-5734 is able to shorten the time to recovery in adults with Covid-19 and lower respiratory tract infection in comparison with placebo. We have addressed this on page 2, lines 9-10 in the revised manuscript.

Reference:

Beigel, J. H.et al., (2020). Remdesivir for the treatment of Covid-19—preliminary report.New England Journal of Medicine.

3. page 2, line 11: Please, comment how often is this the case in immunocompromised patients, e.g., see relevant discussion in Influenza by using appropriate citations such as Eur J Clin Microbiol Infect Dis. 2017 Feb;36(2):361-371. doi: 10.1007/s10096-016-2809- 

Authors’ response:

Different viruses under different antiviral treatments may result in different fitness costs. For examples, it has been shown that in HIV-1, hepatitis B virus and hepatitis C virus, mutations with greater resistance may display higher fitness costs in the absence of selection pressure; however, oseltamivir-resistant H1N1 with mutation H275Y is generally emerged in immunocompromised patients and the resistance mutation caused by antiviral oseltamivir in influenza A virus H1N1 has been found to have little or no fitness costs in the absence of oseltamivir possibly due to compensatory mutations. We have addressed this on page 2, lines14-22 in the revised manuscript.

References:

1.Kossyvakis, A.; Mentis, A. F. A.; Tryfinopoulou, K.; Pogka, V.; Kalliaropoulos, A.; Antalis, E.; Lytras, T.; Meijer, A.; Tsiodras, S.; Karakitsos, P.; Mentis, A. F., Antiviral susceptibility profile of influenza A viruses; keep an eye on immunocompromised patients under prolonged treatment. Eur J Clin Microbiol 2017, 36, (2), 361-371.

2.van der Vries, E.; Stittelaar, K. J.; van Amerongen, G.; Kroeze, E. J. B. V.; de Waal, L.; Fraaij, P. L. A.; Meesters, R. J.; Luider, T. M.; van der Nagel, B.; Koch, B.; Vulto, A. G.; Schutten, M.; Osterhaus, A. D. M. E., Prolonged Influenza Virus Shedding and Emergence of Antiviral Resistance in Immunocompromised Patients and Ferrets. Plos Pathogens 2013, 9, (5).

3. Alonso, M.; Rodriguez-Sanchez, B.; Giannella, M.; Catalan, P.; Gayoso, J.; de Quiros, J. C. L. B.; Bouza, E.; de Viedma, D. G., Resistance and virulence mutations in patients with persistent infection by pandemic 2009 A/H1N1 influenza (vol 50, pg 114, 2011). Journal of Clinical Virology 2011, 51, (2), 150-150.

4. Baz, M.; Abed, Y.; Simon, P.; Hamelin, M. E.; Boivin, G., Effect of the Neuraminidase Mutation H274Y Conferring Resistance to Oseltamivir on the Replicative Capacity and Virulence of Old and Recent Human Influenza A(H1N1) Viruses. Journal of Infectious Diseases 2010, 201, (5), 740-745.

5. Renzette, N.; Caffrey, D. R.; Zeldovich, K. B.; Liu, P.; Gallagher, G. R.; Aiello, D.; Porter, A. J.; Kurt-Jones, E. A.; Bolon, D. N.; Poh, Y. P.; Jensen, J. D.; Schiffer, C. A.; Kowalik, T. F.; Finberg, R. W.; Wang, J. P., Evolution of the Influenza A Virus Genome during Development of Oseltamivir Resistance In Vitro. Journal of Virology 2014, 88, (1), 272-281.

4. page 2, line 19: The reader may wonder to know what happens in adaptive immunity, as well.

Authors’ response:

In adaptive immunity, neutralization antibodies are mainly directed against the viral exposed surface proteins. It has been found that the hemagglutinin genome structure in IAV and glycoprotein E2 gene in HCV genome have been altered in order to escape adaptive immunity. We have addressed this on page 2, lines 30-34 in the revised manuscript.

5. page 2, line 44: Please, elaborate on information about the homology of bovine coronavirus with human coronaviruses.

Authors’ response:

Based on the full-length genome sequence identity, the phylogenetic tree analysis suggests that bovine coronavirus (BCoV) shows higher sequence identity with human coronavirus (HCoVs) HCoV-OC43, HCoV-HKU1, SARS-CoV, MERS-CoV and SARS-CoV-2 (Betacoronavirus) than HCoVs HCoV-229E and HCoV-NL63 (Alphacoronavirus) (see Fig. 1 in the reference below). We have addressed this on page 1, lines 38-41 and page 3, lines 6-11.

Reference:

Chen, B. et al., Overview of lethal human coronaviruses. Signal Transduct Tar 2020, 5, (1).

6. page 3, line 10: Please, explain how this final concentration was selected. 

Authors’ response:

Ten µM GS-5734 was selected based on the study (see Fig. 2 in the reference shown below) in which GS-5734 can inhibit replication of SARS-CoV and MERS-CoV in human airway epithelial cell (HAE) at concentration of 10 µM. We have addressed this on page 3, lines 25-27.

Reference:

Agostini, M. L., Andres, E. L., Sims, A. C., Graham, R. L., Sheahan, T. P., Lu, X., ... & Ray, A. S. (2018). Coronavirus susceptibility to the antiviral remdesivir (GS-5734) is mediated by the viral polymerase and the proofreading exoribonuclease. MBio, 9(2).

7. page 3, line 13: Please, explain how studying innate immunity is possible withinin vitrosettings? Please, also, elaborate on what is meant by “treatment of innate immunity”? So, after having read the Results sections as well, the reviewer would like to request to please explain early on that this can be performed using, for example, stimulants of innate immunity with a double-stranded RNA structure.

Authors’ response:

We thank the reviewer for pointing out the unclear description. We have deleted the inappropriate description “treatment of innate immunity” and explained early in the Methods section that “the innate immunity can be induced using polyinosinic:polycytidylic acid (poly IC), a stimulant of innate immunity with a double-stranded RNA structure.” We have addressed this on page 3, lines 33-37.

8. page 3, line 22: Since the comparison is made between “before” and “after” treatment, why was an unpaired t-test, and not paired t-test applied?

Authors’ response:

We thank the reviewer for pointing out the unclear description regarding the student t test. In Figures 2 and 4, a paired t-test was employed and in figures 3 and 6, an unpaired t-test was used. We have clarified this on page 4, line 43 and specified these in figure legends on page 7, line 22; page 9, line 5; page 11, line 18; page 12, line 24.

9. page 4, line 32: The phrase “[…] performed based on the clinical trial results” appears unsupported from the Methods section. Please, provide explanations

Authors’ response:

We thank the reviewer for pointing out the unclear description. The aim of the sentence is to explain that the reason why 10 times of virus passages were selected for the current study was based on the previous study showing that 10-day dosing durations of the GS-5734 (one dose per day) can shorten the time to recovery in adults infected with SARS-CoV-2. To avoid the misunderstanding, we have deleted the phrase in Results section and clarified the description in Methods section. This has been addressed on page 3, lines 29-31.

10. page 12, line 32 (Optional): Would it be possible to perform these experiments at the authors’ side?

11. page 12, line 21-41 (Optional): This text is rather speculative. Although elegantly written, the reviewer recommend that the authors either “tone it down” or create a figure and state these elements as “hypothesis”, or even better perform some (at least preliminary yet publication-standards in vitroexperiments) to justify their conclusions.

12. page 12, line 21-41: Please, conduct at least some molecular dynamics experiments to justify your claims, following similar examples in the field: https://pubmed.ncbi.nlm.nih.gov/32637944/

Authors’ response to Comments 10, 11 and 12:

We thank the reviewer for the valuable suggestions to improve the manuscript. We agree that we may have over-interpreted the results without substantial evident supported from current study and others. We therefore adopt the reviewer’s suggestion by eliminating over-interpreted claims and toning down the statements. These have been addressed on page 13, lines 7-19. 

13. pages 13-14, “Conclusions” section: Please, “tone down” these statements.

Authors’ response:

We thank the reviewer for the suggestion and have toned down the statements in Conclusion section based on the experimental results. We have addressed this on page 14, lines 31-38.

14. page 13, line 30: Please, comment on whether there are experiments to assess your findings in connection to the adaptive immunity.

Authors’ response:

We have provided experimental evidence (see references below) to connect the current findings to the adaptive immunity. We have addressed this on page 14, lines 7-12.   

References:

1. McKinley, E. T.; Hilt, D. A.; Jackwood, M. W., Avian coronavirus infectious bronchitis attenuated live vaccines undergo selection of subpopulations and mutations following vaccination. Vaccine 2008, 26, (10), 1274-1284.

2. Ellis, S.; Keep, S.; Britton, P.; de Wit, S.; Bickerton, E.; Vervelde, L., Recombinant Infectious Bronchitis Viruses Expressing Chimeric Spike Glycoproteins Induce Partial Protective Immunity against Homologous Challenge despite Limited Replication In Vivo. Journal of Virology 2018, 92, (23).

3. Cavanagh, D.; Ellis, M. M.; Cook, J. K. A., Relationship between sequence variation in the S1 spike protein of infectious bronchitis virus and the extent of cross-protection in vivo. Avian Pathology 1997, 26, (1), 63-74.

4. Keep, S.; Sives, S.; Stevenson-Leggett, P.; Britton, P.; Vervelde, L.; Bickerton, E., Limited Cross-Protection against Infectious Bronchitis Provided by Recombinant Infectious Bronchitis Viruses Expressing Heterologous Spike Glycoproteins. Vaccines-Basel 2020, 8, (2)

Minor comments:

1. page 3, line 8: Please, provide explanation of the acronym MOI.

Authors’ response: Multiplicity of infection (MOI) indicates the number of virus particles infecting one cell. We have addressed this on page 3, line 24.

2. page 4. line 22: Please, provide the name of the city for the company that produces the software.

Authors’ response: We have addressed this on page 4, line 44.

3. page 7. line 16: Please, correct typo to “alternative host cells”

Authors’ response: We have addressed this on page 7, line 26.

4. page 11.line 21: Please, correct typo to “develop”.

Authors’ response: We have addressed this on page 12, line 6.

5. page 13, line 28: Please, insert commas as follows: “capability, and, thus, the viral fitness …”

Authors’ response: We have addressed this on page 14, line 5.

6. page 13, line 38: Please, correct typo to “evolutionarily”.

Authors’ response: We have addressed this on page 14, line 18.

7. General comment: Please, do not separate the Figures from their legends in separate lines, as the reader’s attention is heavily dispersed.

Authors’ response: We have deleted the line between figures and their legends. These have been modified on pages 5, 7, 8, 9, 11 and 12.

Sincerely,

Hung-Yi Wu, D.V.M., M.S., Ph.D.

Professor of Graduate Institute of Veterinary Pathobiology

College of Veterinary Medicine

National Chung Hsing University

Tel:886-4-22840369

Fax:886-4-22862073 ;Email:[email protected]